# ASSESSING TWO NOVEL DISTANCE-BASED LOSS FUNCTIONS FOR FEW-SHOT IMAGE CLASSIFICATION

## ABSTRACT

Few-shot learning is a challenging area of research which aims to learn new concepts with only a few labeled samples of data. Recent works based on metric-learning approaches benefit from the meta-learning process in which we have episodic tasks conformed by support set (training) and query set (test), and the objective is to learn a similarity comparison metric between those sets. Due to the lack of data, the learning process of the embedding network becomes an important part of the few-shot task. In this work, we propose two different loss functions which consider the importance of the embedding vectors by looking at the intra-class and inter-class distance between the few data. The first loss function is the Proto-Triplet Loss, which is based on the original triplet loss with the modifications needed to better work on few-shot scenarios. The second loss function is based on an inter and intra class nearest neighbors score, which help us to know the quality of embeddings obtained from the trained network. Extensive experimental results on the miniImagenNet benchmark increase the accuracy performance from other metric-based few-shot learning methods by a margin of 2%, demonstrating the capability of these loss functions to allow the network to generalize better to previously unseen classes.

## 1 INTRODUCTION

Despite the advances in deep learning research, it remains a challenge for the standard supervised learning to achieve satisfactory results when learning from just a small amount of labeled data. Current deep learning algorithms tend to overfit when they are given a small dataset for training, reducing their generalization capabilities. Moreover, there are many problem domains where obtaining labeled data can be very difficult or imply a lot of manual work to get the data with its ground truth, representing a problem for real world applications as it is time consuming and costly.

Few-shot learning (FSL) methods has been proposed (Koch et al. (2015); Snell et al. (2017); Vinyals et al. (2016); Sung et al. (2017); Finn et al. (2017); Nichol et al. (2018)) to classify previously unseen data into a set of new classes, given only a small amount of labeled instances per class. The main challenge for FSL is to apply a fine-tuning process to an existing embedding network to adapt to new classes, with the problem that this could easily lead to overfitting due to the few labeled samples available for each class. There are two main FSL approaches: The first one is Meta-learning based methods (Finn et al. (2017); Andrychowicz et al. (2016); Li & Malik (2016); Chen et al. (2017)), where the basic idea is to learn from diverse tasks and datasets and adapt the learned algorithm to novel datasets. The second are Metric-learning based methods (Xing et al. (2003); Koch et al. (2015)), where the objective is to learn a pairwise similarity metric such that the score is high for similar samples and dissimilar samples get a low score. Later on, these metric learning methods started to adopt the meta learning policy to learn across tasks (Snell et al. (2017); Vinyals et al. (2016); Sung et al. (2017)). The main objective of these methods is to learn an effective embedding network in order to extract useful features of the task and discriminate on the classes which we are trying to predict. From this basic learning setting, many extensions have been proposed to improve the performance of metric learning methods. Some of these works focus on pre-training the embedding network (Chen et al. (2019a)), others introduce task attention modules (Chen et al. (2020); Li et al. (2019a); Zheng et al. (2019)), whereas other try to optimize the embeddings (Lee et al. (2019)) and yet others try to use a variety of loss functions (Zheng et al. (2019)).

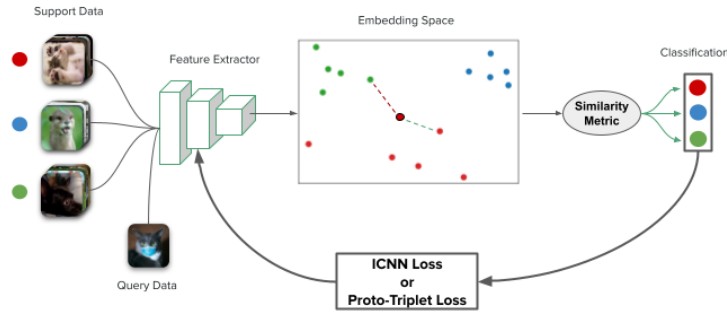

Figure 1: We propose two loss functions that works well for metric meta-learning approaches for few-shot classification. We optimize an embedding network based on the error calculated by the ICNN Loss or the proto-triplet loss, which aims to increase the inter-class distance and decrease the intra-class distance.

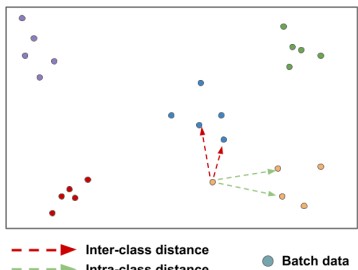

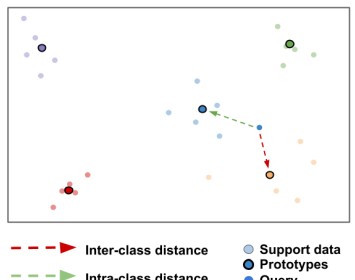

(a) The ICNN Loss is based on a score given by the intra and inter class distance from the batch points. For each data point, we measure its Inter and Intra Class Nearest Neighbors score and optimize the embedding network based on the quality of features.

(b) The Proto-Triplet Loss is based on the triplet loss. This loss takes as anchor a query point, the prototype from the same class as positive point and the nearest prototype of different class as the negative point.

In this work, we propose two different loss functions based on the concepts of inter-class and intra-class distance. As showed in Figure 1, these loss functions allow us to optimize the embedding network and learn more discriminative features across tasks. For one of the proposed loss functions, we take as inspiration one of the most widely used losses in metric-learning: the triplet loss. For the second one, we adopt an algorithm based on nearest neighbors distance. We demonstrate the effectiveness of our loss functions and show our competitive results compared with state-of-the-art methods.

The rest of the paper is organized as follows: In Section 2, we describe the related work in the few-shot learning problem. In Section 3 we described the proposed model. We first explain the Proto-triplet loss function, and then we explain the ICNN Loss function. Then we discuss some of the design choices we needed to take for the ICNN loss. In Section 4 we detail the experimental setup used for the implementation of the models. In Section 5 we show the results obtained and discuss about its performance. Finally, in Section 5 we present our conclusions and discuss the future work.

## 2 MOTIVATION AND RELATED WORK

### 2.1 DEEP METRIC LEARNING

The goal of metric learning is to learn a similarity function from the data. More specifically, it aims to learn feature embeddings in a way that reduce the distance between embeddings corresponding to instances of the same class (intra-class) and increase the distance between embeddings corresponding to instances of different class. Deep metric learning uses an embedding network to learn

the discriminative features that will be used to compute the similarity metric. Below we review the more relevant deep metric learning methods.

One of the fundamental methods for metric learning is the Siamese Networks (Koch et al. (2015)), which is a symmetric neural network architecture that consists on two subnetworks both having the same parameters. These networks learn its parameters by calculating a distance metric between the feature embeddings of each subnetwork each with a different input. The loss function used in Siamese Networks is the contrastive loss or pairwise ranking loss, which seeks for the distance of samples from the same class to be small and from different class to be large.

The second important metric learning method is the Triplet Network (Schroff et al. (2015)), which is also a symmetric neural network architecture but this method consist of three identical subnetworks sharing the same parameters. The input of the three subnetworks consist on three different images: The first one is the anchor (the baseline image), the second is the positive sample (an instance that belongs to the same class as the anchor), and the third is the negative sample (an instance that belongs to a different class than the anchor). This network use the triplet loss to learn discriminative feature embeddings, and it works by ensuring that the anchor image is close to the positive images and far away from the negative images.

These metric learning methods have been widely used for different purposes as image retrieval (Wang et al. (2014)), face recognition (Schroff et al. (2015); Taigman et al. (2014); Hu et al. (2014)), person re-identification (Xiao et al. (2017)), video surveillance (Huang et al. (2018)), three-dimensional modelling (Dai et al. (2017)), signature verification (Bromley et al. (1993)), medical image analysis (Annarumma & Montana (2017)), text understanding (Mueller & Thyagarajan (2016); Benajiba et al. (2018)), among other problems.

## 2.2 Meta-learning for Few-shot learning

As deep learning started to produce good results in many machine learning problems, some works proposed to use the meta-learning policy in order to optimize deep models Andrychowicz et al. (2016); Li & Malik (2016); Chen et al. (2017). The meta-learning policy refers to learn across tasks and then adapt to new tasks, instead of learning to the level of samples. The meta-learning objective is to learn the parameters $\theta$ that minimize the loss across all tasks. Few-shot learning is the perfect process in which we can test meta-learning algorithms, because of the few-labeled data given to each task.

The meta-learning approach to tackle a few-shot learning problem is divided into two stages: meta-train and meta-test. The meta-learning setup consists of episodic tasks, which can be seen as batches in traditional deep learning. A few-shot $K$-way $C$-shot image classification task is given $K$ classes and $C$ images per class. The task-specific dataset can be formulated as $D = \{D_{train}, D_{test}\}$, where $D_{train} = \{(X_i, y_i)\}_{i=1}^{N_{train}}$ denotes the classes reserved for the training phase and $D_{test} = \{(X_i, y_i)\}_{i=1}^{N_{test}}$ denotes the classes reserved for the testing phase. For each meta-train task $T$, $K$ class labels are randomly chosen from $D_{train}$ to form a support set and a query set. The support set, denoted by $S$, contains $K \times C$ samples ($K$-way $C$-shot) and the query set, denoted by $Q$, contains $n$ number of randomly chosen samples from the $K$ classes. The training phase use an episodic mechanism, where each episode $E$ is loaded with a new random task taken from the training data. For the meta-test, the model is tested with a new task $T$ constructed with classes that weren't seen during the meta-train.

We can summarize the few-shot learning methods based on what the model seeks to meta-learn. Some approaches consists on having a base-learner and a meta-learner, where meta-learner parameters are optimized by gradual learn across tasks to facilitate the fast learning of the base-learner for each specific task. MAML (Finn et al. (2017)), is one of these methods and have the idea to search for a good parameter initialization such that the base learner can rapidly generalize with this initialization. Then, REPTILE (Nichol et al. (2018)) incorporates an $L_2$ loss to simplify the computation of MAML. Further on, LEO (Rusu et al. (2018)) is proposed as a network to learn low dimension latent embedding of the model. Meta-SGD (Li et al. (2017)) also learns the base learner update direction and learning rate on the meta-learning process. Meta-Leearner LSTM (Ravi & Larochelle (2017)) propose to finetune the base learner by a LSTM-based meta-learner, which takes as input the loss and gradient of base learner with respect to each support sample. Other approaches seek to learn the similarity metric that is expected to be transferrable across different tasks.

## 2.3 METRIC META-LEARNING FOR FEW-SHOT LEARNING

There is a whole branch of meta-learning approaches to solve the few-shot learning problem by inheriting the main idea of metric learning. These approaches adopts the meta-learning setup to learn the similarity metric expected to generalize across different tasks. There are baseline methods which achieved important milestones for few-shot learning, such as Prototypical Networks (Snell et al. (2017)), Matching Networks (Vinyals et al. (2016)) and Relation Networks (Sung et al. (2017)). Prototypical Networks is the model which we are using as a basis, and it works by taking the center of support samples' embeddings from each class to create the class prototypes. Then, the model use a distance metric (typically the euclidean distance) to predict the probabilities for each query sample. The Matching Networks predicts the probability of query samples by measuring the cosine similarity between the query embedding and each support sample embedding. The Relation Networks adopts a learnable CNN as the pairwse similarity metric, which takes the concatenation of feature maps of support sample and query sample as input and outputs the relation score. These three methods can be considered as the base metric learning approaches for few-shot learning.

Further on, some recent works focus on introducing task attention modules (Chen et al. (2020); Li et al. (2019a); Zheng et al. (2019)), whereas others try to optimize the embeddings (Lee et al. (2019)) and others add a second term to the loss function (Zheng et al. (2019)). Up until now, there is a lack of research for loss functions which work for the problem of few-shot learning tackled from a metric-learning perspective. Our proposed model take part of these meta-learning approaches based on metric learning, by adopting the idea of the intra-class and inter-class variance into two different loss functions, which will help us to better optimize an embedding network to obtain more discriminant feature vectors.

# 3 PROPOSED MODEL

We propose two different loss functions for training a Convolutional Neural Network (CNN) as an embedding function. These loss functions optimize the embedding space across the meta-learning tasks.

## 3.1 PROTO-TRIPLET LOSS

The traditional Triplet Loss (Schroff et al. (2015)) is widely used for metric learning, where the objective is to train a learner by using a similarity comparison metric between the sampled data. The loss function aims at pulling similar samples close to each other, while pushing away the samples of different classes far away. The objective of building the triplets consists on creating reference points for a query (anchor), and obtain the distance between the query and the embedding of an instance in the same class (positive), and the distance between the query and the embedding of an instance in different class (negative).

The Proto-Triplet Loss is based on the original Triplet Loss with the difference that we are using the data points obtained as prototypes for calculating the loss, making it more suitable for the few-shot learning setting. The formula for the proto-triplet loss is similar to the original triplet loss:

$$L(X_a, X_p, X_n) = \left[ \|f_a - f_p\|^2 - \|f_a - f_n\|^2 + \alpha) \right]_+ , \tag{1}$$

where $[]_+$ is the max between 0 and the result, also known as hinge loss. $f_a$ is the embedding from the query, $f_p$ is the prototype from the same class as the query and $f_n$ is the nearest prototype of different class as the query. $\alpha$ is a margin hyper parameter to be enforced between the negative and positive pairs. The goal of this function is to keep the distance between the query and the prototype of same class smaller than the distance between the query and the prototype from a different class.

Furthermore, inspired in (Li et al. (2019b)), we propose to use the Proto-Triplet Loss with $K$ reference points to not only use one negative sample. In the meta-learning setting, we compare an instance with multiple classes, so our loss function should be more effective by performing a comparison with more prototypes of different classes.

For the negative samples, we take the $K$ nearest prototypes of different class to form the triplets. The modified function is formulated as:

$$L(X_a, X_p, X_n) = \frac{1}{K} \sum_{i \in U} \left[ \|f_a - f_p\|^2 - \|f_a - f_n\|^2 + \alpha \right]_+, \tag{2}$$

where $U$ is the set of triplets made by the $K$ negative prototypes. The rest of the terms in the formula are the same as stated in the previous equation.

## 3.2 ICNN Loss

The Inter and Intra Class Nearest Neighbors Score (ICNN Score) (García Ramírez (2021)) was proposed as a measure to remove noisy features and improve the performance of manifold-based algorithms for supervised dimensionality reduction, aiding the feature selection by a subset evaluation. The ICNN is based on two main concepts: the inter-class and the intra-class distances. The former describes the distance between data points of different classes, while the latter refers to the distance between data points of the same class.

The ICNN assigns a score by measuring the distance and variance of the inter and intra k-nearest neighbors of each instance in the data. The equation to measure the data points and assign them a score is represented by three different terms: $\lambda$, $\omega$ and $\gamma$. The ICNN formula is given as follows:

$$ICNN(X) = \frac{1}{|X|} \sum_{x_i \in X} \lambda(X_i)^{\frac{1}{p}} \omega(X_i)^{\frac{1}{q}} \gamma(X_i)^{\frac{1}{r}}, \tag{3}$$

where $p$, $q$ and $r$ are control constants.

$\lambda$ is a function that penalizes the neighbors of $X_i$ with the same class based on how distant they are, and the neighbors of different classes based on how close they are. If the neighbors of different classes are very distant, and the neighbors with same class are very close, the score will be close to 1. The formula for $\lambda$ was modified from the original ICNN proposal (García Ramírez (2021)), to better estimate the score with a few data points scenario. $\lambda$ is calculated as follows:

$$\lambda(X_i) = \frac{\sum_{p \in K x_i} 1 - \frac{d(X_i, p)}{\alpha(X_i)}}{|K_{x_i}|} + \frac{\sum_{q \in K_{\tilde{x}_i}} \frac{d(X_i, q)}{\beta(X_i)}}{|K_{\tilde{x}_i}|}, \tag{4}$$

where $K_{x_i} = KNN(x_i) \in y_i$ are the set of k-nearest neighbors of $x_i$ that have the same class. $K_{\tilde{x}_i} = KNN(x_i) \in y_j \neq y_i$ are the set of k-nearest neighbors of $x_i$ that has different class. $d(a, b)$ is a distance function, which in this case is the euclidean distance. $\alpha(X_i)$ is the distance from $X_i$ to the nearest neighbor of different class. $\beta$ is the maximum distance from $X_i$ to the neighbors of different class. In the ideal scenario, the neighbor's distance of the same class are close to 0 and the distance with different classes are close to 1.

The second function $\omega$ penalizes the distance variance of neighbors (Eq. 5). The distance is calculated as in the $\lambda$ function. A high variance is not desirable as it increases the chance of class overlaps.

$$\omega(X_i) = 1 - \left( Var\left( \sum_{p \in K_{\tilde{x}_i}} \frac{d(X_i, p) - \theta(X_i)}{\alpha(X_i) - \theta(X_i)} \right) + Var\left( \sum_{q \in K_{x_i}} 1 - \frac{d(X_i, q) - \theta(X_i)}{\alpha(X_i) - \theta(X_i)} \right) \right), \tag{5}$$

Lastly, the $\gamma$ function (7) describes the ratio of the neighbor's classes. The optimal situation is when all the neighbors of $X_i$ are in the same class of $X_i$. The formula for $\gamma$ is the following:

$$\gamma(x_i) = \frac{|K_{x_i}|}{|K_{x_i}| + |K_{\tilde{x}_i}|}, \tag{6}$$

where $|K_{x_i}|$ is the number of nearest neighbors in the same class as $X_i$, and $|K_{\tilde{x}_i}|$ is the number of nearest neighbors in different class as $X_i$. Each instance is penalized based on the neighbors in the

same class of $x_i$. Each of the three functions ($\lambda, \omega$ and $\gamma$) have an output that ranges between 0 and 1.

The idea of the ICNN Loss is to use this measure to increase the inter-class distance and reduce the intra-class distance, in order to learn better representations across the meta-tasks. The formula for the ICNN loss is the following:

$$ICNNLoss = -log(ICNN(X)), \tag{7}$$

where we apply a negative log to the ICNN Score to obtain a value close to 0 when the score is close to 1 and a high value when the score is close to 0.

### 3.2.1 DESIGN CHOICES IN ICNN LOSS

Having defined the equation for the ICNN Loss function, we test different scenarios by varying the data points given to the algorithm and combining it with cross entropy loss and proto-triplet loss. For training the network, we know to which class the support data and the query data belongs to, so we can use both sets to assign a score to the current batch. The different options were tested by assigning a different score to the support data, the query data and both the support and query data. For the support data, calculating the ICNN Score is straightforward, as each data point has an assigned score based on its nearest neighbors and averaged across the set. For the query data, each query is given a score based on the nearest neighbors of the support data points. When using both the support and query data, the data points are combined and considered for calculating the score for each instance. The options for the algorithm are the following:

1. Score only on support data
2. Score on support data and score in query data
3. Score of support and query data together

We can also opt to make a different choice of data given to the ICNN algorithm when working with the query set. As our method aims at classifying the queries based on the prototypes, we added the option to give a score to the query set by using the prototypes as neighbors. Each query instance is given a score by using the $k$ nearest prototypes.

## 4 EXPERIMENTAL SETUP

The experiments made on this work were designed to answer the following questions: (1) Is the Proto-Triplet Loss competitive with other metric-learning state-of-the-art models? (2) Is the ICNN Loss competitive with other metric-learning state-of-the-art models? (3) How does our loss functions optimize the feature space to make the features more discriminating?

### 4.1 DATASET

We evaluate our experiments using the MiniImageNet dataset (Vinyals et al. (2016)), which is a version of the ImageNet Large Scale Visual Recognition Competition 2012 (Russakovsky et al. (2014)). Following the split proposed by Ravi & Larochelle (2017), this version of ImageNet is divided into 64 classes for training, 16 classes for validation and 20 classes for testing, making a total of 100 classes for the meta-learning. Each class contains 600 images to have a total of 60,000 images. This dataset is used as a benchmark to evaluate most of the state-of-the-art few-shot learning methods.

### 4.2 EVALUATION METRICS

We follow the same method as other metric learning methods four evaluating our results (Snell et al. (2017); Vinyals et al. (2016); Sung et al. (2017)), reporting the mean accuracy (%) of 1,000 randomly constructed tasks taken from the testing set along with the 95% of confidence interval. Each task of the testing phase contains 15 query images per class.

| Model | Feature Extractor | 1-shot | 5-shot |
|---|---|---|---|
| Matching Networks | ConvNet | $43.56 \pm 0.84\%$ | $55.31 \pm 0.73\%$ |
| Prototypical Networks | ConvNet | $49.42 \pm 0.78\%$ | $68.20 \pm 0.66\%$ |
| Relation Networks | ConvNet | $50.44 \pm 0.82\%$ | $65.32 \pm 0.70\%$ |
| Baseline* | ConvNet | $41.08 \pm 0.70\%$ | $54.50 \pm 0.66\%$ |
| MAML | ConvNet | $48.70 \pm 1.84\%$ | $63.11 \pm 0.92\%$ |
| Reptile | ConvNet | $49.97 \pm 0.32\%$ | $65.99 \pm 0.58\%$ |
| Ours_Proto-Triplet | ConvNet | $\mathbf{49.82 \pm 0.73\%}$ | $\mathbf{68.76 \pm 0.46\%}$ |
| Ours_ICNN | ConvNet | $\mathbf{49.71 \pm 0.78\%}$ | $\mathbf{68.66 \pm 0.58\%}$ |
| SNAIL | ResNet-12 | $55.71 \pm 0.99\%$ | $68.88 \pm 0.92\%$ |
| DN4 | ResNet-12 | $54.37 \pm 0.36\%$ | $74.44 \pm 0.29\%$ |
| TADAM | ResNet-12 | $58.50\%$ | $76.70\%$ |
| K-tuplet Net | ResNet-12 | $58.30 \pm 0.84\%$ | $72.37 \pm 0.63\%$ |
| ProtoNets + CTM | ResNet-12 | $59.34 \pm 0.55\%$ | $77.95 \pm 0.06\%$ |
| Principal Characteristic Net | ResNet-12 | $63.29 \pm 0.76\%$ | $77.08 \pm 0.68\%$ |
| Ours_Proto-Triplet | ResNet-12 | $\mathbf{61.32 \pm 0.52\%}$ | $\mathbf{79.93 \pm 0.76\%}$ |
| Ours_ICNN | ResNet-12 | $\mathbf{60.79 \pm 0.62\%}$ | $\mathbf{80.41 \pm 0.47\%}$ |

Table 1: Test accuracies on miniImagenet in the 5-way setting for both 1-shot and 5-shot

### 4.3 Implementation Details

We follow the same setting as other few-shot learning models (Vinyals et al. (2016); Snell et al. (2017); Sung et al. (2017)), making the experiments under the setting of 5-way 1-shot and 5-way 5-shot, and using 15 query images for each class in the task. The input images are resized to 84 $\times$ 84 and normalized. We construct 100 random tasks for the training phase over 200 epochs. We validate each epoch with another 500 randomly constructed tasks using images from the validation set. In the testing phase we construct 1000 tasks with images from the testing set. We use the same random seed over all the experiments.

We test two different networks for the feature extractor: a ConvNet and a ResNet-12. The ConvNet follows the same architecture setting as previous works (Vinyals et al. (2016), Snell et al. (2017)). This network is composed of 4 layers of convolutional blocks, with each block having a $3 \times 3$ convolution with 64 filters, followed by a batch normalization and a ReLU layer. For this network, we use the Adam Optimizer, as used in previous works, with an initial learning rate of $1 \times 10^{-3}$ and a step size of 20. The ResNet follows the same architecture as other recent works (Li et al. (2019a), Lee et al. (2019)). This network is pre-trained with images from the training set, using the Stochastic Gradient Descent (SGD) optimizer with a momentum of 0.9 and a learning rate of 0.1 over 100 epochs with a batch size of 128. After the pre-training, we meta-train the network using the SGD optimizer with a learning rate of $1 \times 10^{-4}$, momentum of 0.9 and a step size of 20.

## 5 Results and Discussion

### 5.1 Standard few-shot learning evaluation

For the MiniImageNet dataset, we evaluate our method under the two most common few-shot learning settings: 5-way 1-shot and 5-way 5-shot. The results are compared against base metric learning methods for few-shot classification (Snell et al. (2017); Vinyals et al. (2016); Sung et al. (2017)), and against recently proposed methods (Li et al. (2019b;a); Zheng et al. (2019)).

As detailed in Table 1, our method outperforms most of the baselines in the 5-way 1-shot setting, getting only lower accuracy than Reptile (Nichol et al. (2018)) and Relation Networks Sung et al. (2017)) when using a ConvNet as Feature extractor, and lower accuracy than Principal Characteristics Net (Zheng et al. (2019)) when using a ResNet-12. For the 5-way 5-shot setting, our method outperforms all the baseline methods and the recent ones using ConvNet and ResNet-12 as feature extractors.

| Model | Backbone | 1-shot | 5-shot |
|---|---|---|---|
| $(i)$ ICNN in Support | ConvNet | 41.82 | 55.52 |
| $(ii)$ ICNN in Support + Query | ConvNet | 42.81 | 55.93 |
| $(iii)$ CrossEntropy + ICNN in Support | ConvNet | 49.28 | 68.33 |
| $(iv)$ CrossEntropy + ICNN in Support & Query | ConvNet | **49.71** | **68.66** |
| $(v)$ ICNN in Support & Query(Prototypes) | ConvNet | 41.81 | 55.93 |
| $(vi)$ CrossEntropy + ICNN in Support&Query(Protos) | ConvNet | 48.86 | 67.74 |
| $(vii)$ Full ICNN | ConvNet | 45.42 | 65.86 |
| $(viii)$ Cross Entropy + Full ICNN | ConvNet | 48.62 | 68.12 |
| $(i)$ ICNN in Support | ResNet-12 | 56.07 | 77.44 |
| $(ii)$ ICNN in Support & Query | ResNet-12 | 58.93 | 78.42 |
| $(iii)$ CrossEntropy + ICNN in Support | ResNet-12 | 60.55 | 79.11 |
| $(iv)$ CrossEntropy + ICNN in Support & Query | ResNet-12 | 60.50 | 79.58 |
| $(v)$ ICNN in Support + Query(Prototypes) | ResNet-12 | 59.07 | 79.18 |
| $(vi)$ CrossEntropy + ICNN in Support&Query(Protos) | ResNet-12 | **60.79** | **80.41** |
| $(vii)$ Full ICNN | ResNet-12 | 60.37 | 78.79 |
| $(viii)$ Cross Entropy + Full ICNN | ResNet-12 | 60.32 | 79.30 |

Table 2: Design choinces for the ICNN Loss function using a ConvNet and ResNet-12 as feature extractors and 5-way tasks on both 1-shot and 5-shot settings

| Model | Backbone | 1-shot | 5-shot |
|---|---|---|---|
| Proto-Triplet | ConvNet | 48.85 | 67.79 |
| Cross Entropy + Proto-Triplet | ConvNet | 41.66 | 66.09 |
| Proto-Triplet + Full ICNN | ConvNet | 46.00 | 62.72 |
| Cross-Entropy + Proto-Triplet + Full ICNN | ConvNet | **49.82** | **68.76** |
| Proto-Triplet | ResNet-12 | 60.87 | 78.78 |
| Cross Entropy + Proto-Triplet | ResNet-12 | 60.09 | 79.33 |
| Proto-Triplet + Full ICNN | ResNet-12 | 59.58 | 78.53 |
| Cross-Entropy + Proto-Triplet + Full ICNN | ResNet-12 | **61.32** | **79.93** |

Table 3: Design choinces for the Proto-Triplet Loss function using a ConvNet and ResNet-12 as feature extractors and 5-way tasks on both 1-shot and 5-shot settings

When using a ConvNet as the feature extractor, our method achieves an improvement of 8.8% over the baseline (Chen et al. (2019b)), 5.3% over the Matching Networks (Vinyals et al. (2016)) and 0.4% over the base prototypical Networks for the 5-way 1-shot setting. For the setting of 5-way 5-shot, we achieve an improvement of 13.4% over the Matching Networks, 3.4% over the Relation Networks and 0.5% over the prototypical networks.

When using a ResNet-12 as the feature extractor, our method achieves an improvement of 5.6% over SNAIL (Mishra et al. (2018)), 3% over the K-tuplet Net (Li et al. (2019b)) and 2% over the Category Traversal Module (Li et al. (2019a)) for the few-shot setting of 5-way 1-shot. On the setting of 5-way 5-shot, we obtained an improvement of 11.6% over SNAIL, 8.1% over the K-tuplets Net and 3.4% over the Principal Characteristics Net (Zheng et al. (2019)).

## 5.2 ABLATION STUDIES

The ablation studies are split into three parts: the proto-triplets loss experiments, the ICNN loss experiments and experiments with the networks used as feature extractor (ConvNet and ResNet-12). For all the experiments, we repeated the design choices using both networks to compare against baselines and recent methods in few-shot image classification. The models using a ResNet-12 greatly improve the accuracy performance compared to those using a ConvNet.

The results for the design choices for the ICNN loss are reported in table 2. Our baseline is the simplest form of using the ICNN loss, by applying the score only to the support data. This gives us

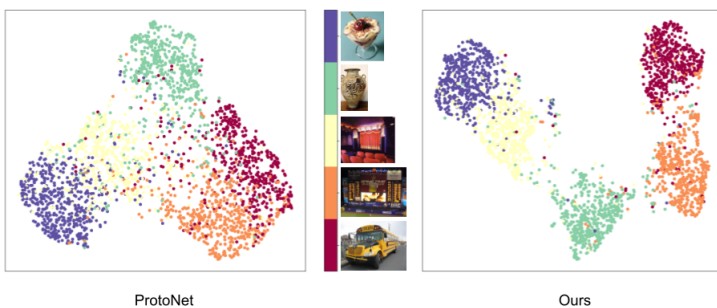

Figure 3: Umap visualization of the improved feature embeddings obtained from the test set. The feature embeddings were computed in a 5-way 5-shot setting by constructing 1,000 tasks from the test data. When creating the tasks for the feature visualization, we kept fix the 5 classes sampled and randomly draw samples to construct the support and query sets

the worst results from the experiments made, and are not competitive with state-of-the-art models. Then, we test our loss function by applying the score to different sets of data. For $(ii)$ and $(iv)$, we test using ICNN on support data and query data separately, with the difference that $(iv)$ uses the additional loss of the cross-entropy. When using a ConvNet, we obtained the best results by using the cross entropy and ICNN in support and query (iv), improving the baseline performance by around 8%. For $(v)$ and $(vi)$ we modify the ICNN score on the query by comparing distances from each query to the prototypes instead of the comparing the distances from each query to the support data. For the case of using a ResNet-12 as the feature extractor, this combination $(vi)$ gave us the best results, improving the baseline by around 4% for the 1-shot setting and 3% for the 5-shot setting. The last experiments, $(vii)$ and $(viii)$, apply the ICNN score to the combined set of support data and query data. These last experiments didn't lead us to the best results, but they are also quite competitive.

The results for the design choices for the proto-triplet loss are reported in table 3. We evaluate the combination of three loss functions: the cross-entropy, proto-triplet and the full version of the ICNN loss. First we test the proto-triplet loss alone to optimize the network, and obtain already a good accuracy performance compared with state-of-the-art results. Then, we start combining the loss functions and we can observe that combining cross entropy with proto-triplet and ICNN loss with proto-triplet we get a worse performance than the obtained using only the proto-triplet loss. The last design choice is the combination of cross entropy, ICNN loss and proto-triplet loss, which is the one with the best accuracy performance by improving the result by 1% using both feature extractors.

## 6   CONCLUSION

In this paper, we proposed two different loss functions to train an embedding network for few-shot image classification. These loss functions are based on the concepts of intra-class and inter-class distance, and have as the main objective to pull together instances of the same class and push further away the instances of different class. The proposed model improves the accuracy performance when compared with baseline models which make use of metric learning approaches to solve the few-shot classification problem on the public benchmark dataset. The proposed model also obtain competitive results when compared with more recent methods which make use of a more robust embedding network, as we improve the accuracy for the 5-way 5-shot setting. Our current framework can be extended in several ways. For instance, we could make the hyper-parameters of the ICNN algorithm to be learnable instead of making them of a fixed value. Another direction is to test these loss functions with different metric meta-learning methods to see if they are allowing the network to learn better feature representations for few-shot tasks.

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

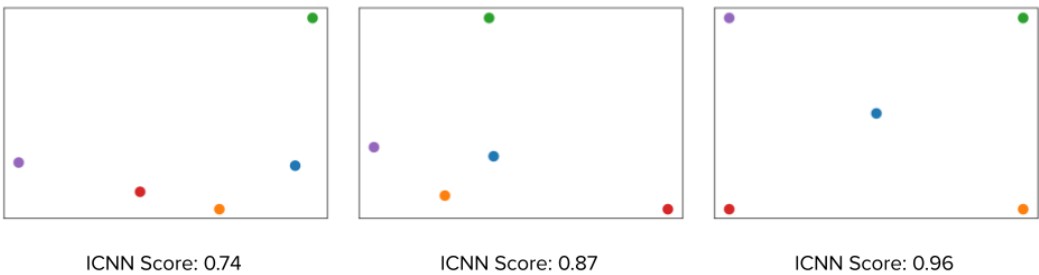

Figure 4: ICNN score behaviour on different 5-way 1-shot scenarios.

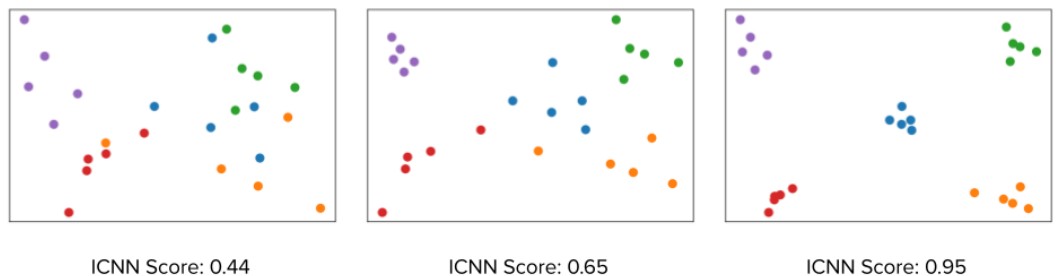

Figure 5: ICNN score behaviour on different 5-way 5-shot scenarios.

## A  APPENDIX

In figure 4 and 5, we can visualize how some few-shot tasks scenarios would be rated using the ICNN score.

