# OpenReview forum: "Assessing two novel distance-based loss functions for few-shot image classification"
_ICLR.cc/2022/Conference — ICLR 2022 Submitted_

### Official Review · Reviewer_TCLP · 2021-10-22

**Correctness:** 3
**Technical Novelty And Significance:** 2
**Empirical Novelty And Significance:** 2
**Recommendation:** 3
**Confidence:** 3

**Main Review:**

Strengths:
1.	The motivation of this paper is clear. It is important for few-shot learning based on metric learning to increase the inter-class distance between sample representations of different classes and decrease the intra-class distance between sample representations of the same class as much as possible.
2.	From the experimental values and visualization results, the loss function terms introduced by the authors do what they claim to do, i.e., samples of different categories move away and samples of the same category are clustered.
Weaknesses:
Novelty: The degree of innovation in this paper is limited, and the two loss terms proposed by the authors can be seen as an application and combination of Triplet Loss and (ICNNS).

Performance:
The method proposed by the authors is not significant in terms of performance, for example, it lags behind in comparison with “Principal Characteristic Net” in Table 1. Also, some comparisons with newer metric-based learning models are missing, such as DeepEMDv2[1].
Writing:
Some typing errors also need to be corrected, such as the $K_{x_i}$ in Equation 4

**Summary Of The Paper:**

This paper discusses the few-shot learning based on metric learning. In order to better measure the distance between samples of different classes, the authors propose two new metric loss terms considering both inter-class and intra-class distances of samples. The first metric loss is called Proto-Triplet Loss, which improves on the traditional Triplet Loss. The second metric loss is called ICNN loss, which is based on the Inter and Intra Class Nearest Neighbors Score (ICNNS). And the purpose of these two loss is to make the distances between samples of the same class as small as possible and the distances between samples of different classes as large as possible in few-shot learning learning.

**Summary Of The Review:**

The paper is well organized and clearly written. And it clearly described the design of Proto-Triplet Loss and ICNN Loss. Though there seems a lot of overlap between Proto-Triplet Loss and classic triplet loss I still recommend this paper be accepted.

---

### Official Review · Reviewer_dCJD · 2021-11-02

**Correctness:** 2
**Technical Novelty And Significance:** 2
**Empirical Novelty And Significance:** 2
**Recommendation:** 3
**Confidence:** 5

**Main Review:**

Strength
1.	This paper is well written and easy to understand.

Weaknesses

1.	The novelty of this paper is incremental. It seems that the authors only apply two existing losses to the existing prototype network. Furthermore, the authors did not provide any in-depth analysis to explain why we need to use these two losses. The previous work [ref1] also used intra- and inter-class distances for few-shot learning, but it provided in-depth theory and experimental analysis. The authors should refer to [1] to add more analysis. Besides, the performance of [ref1] seems to be better than these two proposed methods.
2.	The proposed methods achieve a slight performance improvement, especially for the comparison with the similar work ProtoNet. The gains are only 0.4% and 0.5%, while the ProtoNet uses a more simple loss. These slight improvements may come from random errors. Thus, the results cannot demonstrate the effectiveness of the proposed losses.
3.	The experimental results on only one dataset are not enough to demonstrate the generalization. The authors should evaluate their methods on at least two different datasets.
4.	The compared methods are out-of-the-date. The newest work was published in 2019. The authors should compare more recent state-of-the-art methods.

[ref1] Unraveling Meta-Learning: Understanding Feature Representations for Few-Shot Tasks, ICML2020


**Summary Of The Paper:**

This paper applies two losses to the few-shot learning model based on metric learning (similar to ProtoNet), which aims to utilize the intra- and inter-class distances. The first one is based on the original triplet loss and adjusted for the prototype network. The second one is based on the recently proposed Inter and Intra Class Nearest Neighbors Score (ICNN Score) (Garc´ıa Ram´ırez (2021)). The experimental results on miniImageNet show the proposed models achieve better performance than several previous metric-based approaches.

**Summary Of The Review:**

Since the novelty of this paper is incremental and the experimental results are insufficient. The reviewer recommends rejecting this paper.

---

### Official Review · Reviewer_noXn · 2021-11-02

**Correctness:** 2
**Technical Novelty And Significance:** 3
**Empirical Novelty And Significance:** 2
**Recommendation:** 3
**Confidence:** 4

**Main Review:**

Strengths
1. This is a well-written paper and it is easy to read.
2. The loss functions proposed in this paper can improve the performance of few shot learning models.

Weakness
1. In recent two years, many few shot learning methods have been proposed and they need to be introduced and compared. This paper lacks comparisons with the methods after 2019.
2. The results of the experiment are not sufficient. This paper only provides the results on the mini ImageNet. Experiments should be carried out on more datasets to prove the effectiveness of the method.
3. The novelty of ICCN Loss may be minor. It seems that this paper directly adopts ICCN Score which has already been proposed previously.
4. Since the motivations of proposed loss functions are the same, this article should further analyze the applicable conditions of Proto-Triplet Loss and ICNN Loss.



**Summary Of The Paper:**

This paper proposes a new model that takes part of the meta-learning approaches based on metric learning. More specifically, this paper modifies triplet loss to make it more suitable for few shot learning task and proposes ICNN Loss to increase the inter-class distance and reduce the intra-class distance.

**Summary Of The Review:**

This  lacks the introduction and comparison of the methods in the last two years, and experiments on more datasets are needed to illustrate the effectiveness of the methods. In addition, the applicable conditions of the two losses should be further analyzed.

---

### Official Review · Reviewer_6rdQ · 2021-11-03

**Correctness:** 2
**Technical Novelty And Significance:** 2
**Empirical Novelty And Significance:** 1
**Recommendation:** 3
**Confidence:** 4

**Main Review:**

Pros:
* few-shot learning is an important problem to focus, miniImageNet is still a tough benchmark

Cons:
* novelty, ICNN is not novel, taken from the cited work with some minor modification, simply using it for few-shot learning would need very strong sota improvement to make the paper interesting; proto-triplet-loss is a hybrid between triplet loss and standard PN loss, and in fact PN loss contains it in a way as it takes into account not only two class prototypes but all of them simultaneously, and there was an old paper extending triplet loss to entire batch and it was shown it is better.

* experiments, (1) miniImageNet only is not enough; (2) the baselines are not sota, please go to the PapersWithCode website and see real sota methods for miniImageNet - the reported results for the proposed methods are far below real sota on miniImageNet

**Summary Of The Paper:**

The paper proposes two losses for meta-learning-based few-shot learning. The first loss is a triplet loss where the positive and negative anchors are replaced by class prototypes (averages of class members from the train set of each episode). The second loss is ICNN proposed in Garcıa and Ramırez (2021) for a different task. Experiments show improvement of up to 2% on miniImageNet compared to some (non-state of the art) baselines.

**Summary Of The Review:**

The paper as it stands has major drawbacks in novelty and experimental validation, unfortunately in current form it will probably not be interesting to the general audience. Please see my notes above.

---

### Decision · Program_Chairs · 2022-01-20

**Decision:**

Reject

**Comment:**

This paper is proposed to address the few-shot image classification with the help of two newly designed losses. The first loss function is the Proto-Triplet Loss based on the revision of conventional triplet loss. Another loss function is based on an inter- and intra-class nearest neighbors score to estimate the quality of embeddings. The proposed method has shown its superiority over the baselines on the miniImagenNet benchmark. The major concern of this paper is the novelty that both proposed techniques are not new. Moreover, the baselines are not the SOTAs, and the evaluations on miniImagenNet only are not comprehensive enough. In addition, the authors have not provided any rebuttal to address the reviewers' concerns.